# Matrin3 (MATR3) Expression Is Associated with Hemophagocytosis

**DOI:** 10.3390/biomedicines10092161

**Published:** 2022-09-01

**Authors:** Wen-Chi Yang, Sheng-Fung Lin, Shih-Chi Wu, Chih-Wen Shu

**Affiliations:** 1Division of Hematology and Medical Oncology, Department of Internal Medicine, E-DA Hospital, Kaohsiung 824, Taiwan; 2Faculty of School of Medicine, College of Medicine, I-Shou University, Kaohsiung 824, Taiwan; 3School of Medicine, China Medical University, Taichung 404, Taiwan; 4Trauma and Emergency Center, China Medical University Hospital, Taichung 404, Taiwan; 5Institute of BioPharmaceutical Sciences, National Sun Yat-sen University, Kaohsiung 804, Taiwan

**Keywords:** hemophagocytosis, NF-κB, MATR3, Epstein–Barr virus, necrosis, latex beads

## Abstract

Hemophagocytic lymphohistiocytosis (HLH) is a life-threatening hyperinflammatory syndrome characterized by prolonged fever, cytopenia, hepatosplenomegaly, and hemophagocytosis. This occurs as a result of activated macrophages and impaired function of natural killer cells and/or cytotoxic T lymphocytes. The NF-κB pathway plays a crucial role in hyperinflammation. Matrin3 (MATR3) is a nuclear RNA/DNA-binding protein that plays multiple roles in the regulation of gene expression. We enroll 62 patients diagnosed with secondary HLH and hemophagocytosis. Peripheral blood (PB) from 25 patients and 30 healthy volunteers and good quality bone marrow (BM) samples from 47 patients are collected and used for analysis. Clinical parameters, including age, sex, etiology, ferritin, fibrinogen, triglyceride, and viral infection status, had no association with survival prediction. Patients with downregulation of *NF-κB* and *MATR3*mRNA expression in the BM had a higher mortality rate. *MATR3*mRNA expression in PB was lower in patients compared to that in healthy volunteers. We use *shRNA-MATR3*-KD-THP1 cells to determine the efficacy of phagocytosis. We note that *shRNA-MATR3*-KD-THP1 cells had a higher phagocytic effect on necrotic Jurkat E6 cells and carboxylate modified polystyrene latex beads. Herein, we provide evidence of a new marker for clinical translation that can serve as a potential treatment target for secondary HLH.

## 1. Introduction

Hemophagocytic syndromes (hemophagocytic lymphohistiocytosis, HLH) represent a severe hyperinflammatory condition with cardinal symptoms including prolonged fever, cytopenia, hepatosplenomegaly, and hemophagocytosis. Secondary HLH is caused as a result of activated macrophages and impaired natural killer (NK) cells and/or cytotoxic T lymphocytes. Biochemical indicators of HLH include elevated ferritin and triglycerides and low fibrinogen [1,2,3,4]. HLH has been categorized into primary/familial HLH and reactive/secondary HLH. Primary HLH occurs in early childhood and is driven by homozygosity or compound heterozygosity of mutations in genes affecting cytotoxic cell functions [5,6]. Secondary HLH affects people of all ages and is more frequent in older children and adults who present with no known genetic etiology. Up to 40% of secondary HLH cases occur in adults, with the median age ranging between the mid-40s and 50s. Infections, mainly Epstein–Barr virus (EBV) and human immunodeficiency virus infections, are the primary etiology of secondary HLH in adults, followed by malignancies, such as B cell and NK/T cell lymphomas, and autoimmune diseases primarily systemic lupus erythematosus (SLE) [7]. The prognosis of secondary HLH is poor, and the mortality rate remains high and ranges from 8 to 22% in rheumatologic HLH to 18–24% in EBV-related HLH [8].

The symptoms of HLH are driven by aberrantly high concentrations of inflammatory cytokines, including interleukin (IL)-1, IL-6, IL-10, IL-12, IL-16, IL-18, tumor necrosis factor (TNF)-α, and interferon (IFN)-γ, which result in the activation of dendritic cells, macrophages, NK cells, and CD8+ T cells [9]. Inflammatory cytokines can also lead to the activation of nuclear factor-κB (NF-κB), which has a crucial role in innate and adaptive immune responses [10]. EBV latent membrane protein-1 (LMP-1) is the viral product responsible for the activation of the TNF receptor (TNFR) associated factors/NF-κB/ERK pathway and enhances cytokine secretion via the suppression of the SAP/SH2D1A gene. The activation of NF-κB protects against TNF-α-induced apoptosis of EBV-infected T cells through the downregulation of TNFR-1 [11,12].

Matrin-3 (MATR-3) encodes a nuclear matrix protein and is proposed to stabilize certain messenger RNA species. It plays a role in the regulation of DNA virus-mediated innate immune response by assembling into the HEXIM1-DNA-PK-paraspeckle components-ribonucleoprotein complex (HDP-RNP complex), a complex that serves as a platform for interferon regulatory factor 3 (IRF3) phosphorylation and subsequent innate immune response activation through the cyclic-GMP-AMP (cGAMP) synthase-stimulator of interferon genes (cGAS-STING) pathway [13]. EBV LMP1 TES2 downregulates two RNAs (MATR3 and CGA), which are related to the NFκB pathway [14]. In our study, we analyzed the significance of NF-κB and MATR3 genes in adult patients with secondary HLH of various etiologies.

## 2. Materials and Methods

### 2.1. Patient Recruitment

We enrolled 62 patients who had a confirmed diagnosis of hemophagoctyosis in the bone marrow (BM) and presented with clinical symptoms such as fever, organomegaly, hyperferritinemia, hypertriglyceridemia, and related hypofibrinogenemia, as well as 30 healthy volunteers at Kaohsiung Medical University, Chung-Ho Memorial Hospital, Taiwan, from 2011 to 2015. Peripheral blood (PB) was obtained from 25 patients and 30 healthy volunteers. Mononuclear cells were isolated using Ficoll Paque (Sigma-Aldrich, St. Louis, MO, USA). BM samples of good quality from 47 patients were subsequently used for RNA extraction. The protocol was approved by the Institutional Review Board of the Kaohsiung Medical University Hospital (KMUH-IRB-990410).

### 2.2. Real-Time qRT-PCR to Quantify MATR3 and NF-κB Expression

Total RNA was extracted from the BM samples of the enrolled patients and *MATR3*-knockdown (KD) THP1 cells using Trizol (Invitrogen, Life Technologies, Waltham, MA, USA). *MATR3* mRNA and *NF-κB* mRNA expression were evaluated by quantitative reverse-transcriptase polymerase chain reaction (qRT-PCR) and the specific forward and reverse primers for the TaqMan^®^ probe were designed using Primer Express software version 1.5 (Applied Biosystems, Life Technologies). β-actin was used as an internal control.

### 2.3. Clinical Endpoint

The endpoint for the follow-up of patients was the date of death, and patients who were lost to follow-up were denoted as “censored” data. Overall survival (OS) was defined as the time measured from the date of initial diagnosis until the date of death of the patient; this was denoted as “censored data” for patients who were alive till the last follow-up.

### 2.4. RNA Interference-Mediated MATR3-Knockdown in THP1 Cells and Cell Culture

The shRNA-*MATR3* lentivirus was purchased from Sigma Aldrich (St. Louis, MO, USA). The clones TRCN0000074903, TRCN0000074905, TRCN00000293553, TRCN00000293617, and TRCN00000293618 were identified as *shRNA-MATR3-1*, *shRNA-MATR3-2*, *shRNA-MATR3-3*, *shRNA-MATR3-4*, and *shRNA-MATR3-5*, respectively. Naive THP1 acute myelomonocytic leukemia cells were transfected with lentivirus-expressing shRNAs. Subsequently, the transfected THP1 cells were cultured in a complete RPMI medium. Puromycin (1 µg/mL) was added for the stress selection of THP1 cells infected with shRNA (PLKO-1) empty vector and *shRNA-MATR3* lentivirus (Sigma, St. Louis, MO, USA, Supplementary Methods). The efficacy of *MATR3*-KD was assessed by qRT-PCR.

### 2.5. Phagocytosis of Necrotic and Apoptotic Cells

Jurkat E6 cells were cultured in the same complete medium as THP1 cells at 37 °C in a humidified atmosphere containing 5% CO_2._

Incubation of Jurkat E6 cells at 56° for 30 min would induce apoptosis (Appendix A) [15,16], whereas culturing Jurkat E6 at −80 °C for 30 min would induce necrosis [16]. We cultured three groups of 1 × 10^6^ Jurkat E6 cells at 56 °C for 30 min, at −80 °C for 30 min, and at 37 °C for 30 min, respectively. Then we cultured the three different groups of cells with THP1 complete RPMI medium separately, in 96-well platelet and co-culture with 1 × 10^5^ *shRNA-MATR3-2*-KD THP1 cells, *shRNA-MATR3-4*-KD THP1 cells, shRNA empty vector-transfected THP1 cells, or naïve THP1 cells, respectively and incubated them at 37 °C in a humidified atmosphere containing 5% CO_2_ for 3 h. The cells were collected from each well, washed twice with cold PBS, resuspended in 100 µL of PBS, and stained with 20 µL of CD3 monoclonal Ab (IM1280, Beckman Coulter, Brea, CA, USA) without fluorescence at room temperature for 20 min (each tube contained 2 × 10^5^ THP1 and 2 × 10^6^ Jurkat E6 cells), to determine the surface expression of CD3. Subsequently, the cells were washed twice with cold PBS and 300 µL of PBS was added. Then we used CD45, CD11b, or CD13 monoclonal Abs to stain cells at room temperature for 20 min. IntrpPrepTM permeabilization reagent (A07802, Beckman Coulter, Brea, CA, USA) was then added to allow cyto-CD3-FITC monoclonal Ab to penetrate the cell membrane and stain CD3 (+) Jurkat E6 cells within THP1 cells (THP1 cells do not have CD3 marker). We then used flow cytometry to analyze the expression of cyto-CD3 in THP1 cells.

### 2.6. Hemophagocytosis

We added 1 × 10^5^ *shRNA-MATR3-2*-KD THP1 cells, *shRNA-MATR3-4*-KD THP1 cells, shRNA empty vector-transfected THP1 cells, and naïve THP1 cells in a 96 well plate and cultured them with complete medium for 1 h. Subsequently, 0.1 µL (contains 1 × 10^6^ beads) and 0.5 µL latex beads (L3030, carboxylate modified polystyrene, fluorescent red, Sigma-Aldrich, MO, USA) with red fluorescence were added to separate wells and cultured for 1 h and 3 h, respectively, at 37 °C in a humidified atmosphere containing 5% CO_2_ [17]. THP1 surface cells and latex beads were washed twice with cold PBS, re-suspended in 100 µL PBS, and stained with 100 µL/mL CD33-FITC antibody for 30 min, and flow cytometry was used to determine CD33 expression. The relative amount of ingested PE-latex beads was calculated by subtracting the mean fluorescence intensity of THP1 cells alone from that of each test sample [18].

### 2.7. Flow Cytometry for Determining Phagocytosis

Flow cytometry was performed using a Gallios™ flow cytometer (BeckmanCoulter, Brea, CA, USA). Annexin V assay was used to detect the apoptosis rate of Jurkat E6 cells, as described previously [19]. The expression of surface markers, including CD3, CD13, CD33, CD7, CD11b, and CD14 in THP1 and Jurkat E6 cells was analyzed. Cytoplasmic CD3 and CD13 expressions were analyzed to determine phagocytosis. CD33-FITC and PE for latex beads were analyzed using flow cytometry.

### 2.8. Fluorescent Microscope

The phagocytosis of the latex beads of THP1 cells was observed under a fluorescent microscope using Research Inverted system IX73 (Olympus).

### 2.9. Statistics

Statistical analyses were performed using SPSS 17.0 (IBM SPSS software, Armonk, NY, USA). Differences in age; clinical laboratory data, including ferritin, fibrinogen, triglyceride (TG), lactate dehydrogenase (LDH), β2-microglobulin, white blood cells, hemoglobulin, platelet, and PB mononuclear cells; *MATR3*mRNA and *NF-κB* mRNA expressions in the BM between diagnosis and death/survival were analyzed using one-way analysis of variance (ANOVA) test. ANOVA test was used to determine the differences in MATR3mRNA and NF-κB mRNA expressions in different etiologies of HLH. Differences in etiology and viral infection status (EBV, cytomegalovirus (CMV), and Varicella-Zoster Virus (VZV)) between death and survival were analyzed using the Chi-squared test. Correlation regression analysis was used to assess the correlations among *MATR3*mRNA, *NF-κB* mRNA expressions, and ferritin levels. Kaplan–Meier curves were plotted to estimate survival.

## 3. Results

### 3.1. Patients’ Characteristics

The patients’ characteristics are shown in Table 1. There was no difference between clinical data, including age, sex, ferritin, triglyceride, fibrinogen, cytopenia status, LDH, β2-microglobulin, and viral infection status (EBV, CMV, and VZV) between the deceased patients and those who survived. We found that patients with lower expression of *NFκB* and *MATR3* in the BM had a higher risk of mortality (*p* = 0.011, and *p* = 0.01, respectively, Table 1).

There was no association between the mortality rate and the different etiologies (Table 2). However, patients with infection-induced hemophagocytosis had the best OS, other than that of patients with a hereditary etiology (Figure 1). However, the association could have possibly been seen owing to the small sample size and the diverse etiologies.

### 3.2. Downregulation of MATR3 in Patients with Hemophagocytosis

We analyzed *NFκB* and *MATR3* mRNA expressions in the PB mononuclear cells of patients with hemophagocytosis and healthy volunteers. Because both genes are long, we designed 2 sets of primers for the detection of each gene. *MATR3* mRNA was downregulated in PB mononuclear cells of patients compared to that in healthy volunteers. (Figure 2C,D). However, there was no significant difference in *NFκB* mRNA expression between that of patients and healthy volunteers (Figure 2A,B).

However, there was no significant correlation in *NFκB* and MATR3mRNA expression between PB mononuclear cells and BM cells (Appendix A). We observed a positive correlation between *NFκB* and *MATR3*mRNA expressions in BM cells, as well as a positive correlation between *NFκB* and *MATR3*mRNA expressions in PB mononuclear cells (Appendix A). Correlation analysis of clinical factors, including laboratory data, EBV status, as well as *NFκB* and *MATR3*mRNA expression of PBMCs and BM cells revealed a positive correlation only between *MATR3*mRNA expression in PB mononuclear cells and platelet count at diagnosis (Pearson correlation coefficient = −0.685 *p* = 0.003, data not shown), as well as a positive correlation between LDH and TG (Pearson correlation coefficient = 0.441, *p* = 0.021, data not shown).

### 3.3. Similar NFκB and MATR3mRNA Expression in Different Disease Etiologies

In our patient cohort, we did not observe any differences in *NFκB* and *MATR3* mRNA expression in both PB and BM cells of patients with different etiologies (Table 3).

### 3.4. Enhanced Phagocytosis in shRNA-MATR3-KD THP1 Cells

We generated *shRNAMATR3*KD THP1 cells to analyze the phagocytic efficacy. The *shRNA4 MATR3* KD THP1 cells had lower MATR3 mRNA expression, followed by *shRNA3* and *shRNA2*
*MATR3* KD THP1 cells (Figure 3). There is no correlation between *MATR3* and *NF**κB* mRNA expression in *MATR3*-KD THP1 cells.

#### 3.4.1. Increased Phagocytosis of Latex Beads in shRNA-MATR3-KD THP1 Cells

We used 0.1 μL and 0.5 μL carboxylate modified polystyrene latex beads, with red fluorescent, as a phagocytosis stimulator to evaluate the phagocytotic ability of *shRNA-MATR3*-KD THP1 cells. Co-culture of *shRNA-MATR3*-KD THP1 cells with 0.5 μL latex beads and THP1 cells for 3 h showed better phagocytic effects. More latex beads were phagocytosed by *shRNA2-MATR3*-KD and *shRNA4-MATR3*-KD THP1 cells, compared to that by the control vector and naïve THP1 cells, based on flow cytometry and fluorescent microscope analysis (Figure 4, Appendix A).

#### 3.4.2. Phagocytosis of Necrotic Jurkat E6 Cells Increase in shRNA-MATR3-KD THP1 Cells

We observed that Jurkat E6 cells express CD3, but not CD13 or CD11b, whereas THP1 cells express CD13 and CD11b, but not CD3. After treating healthy Jurkat E6 cells at 80 °C for 30 min, we co-cultured treated Jurkat E6 cells with *shRNA2-MATR3* KD THP1 cells, *shRNA4-MATR3* KD THP1 cells, control vector-transfected THP1 cells, and naïve THP1 cells at 37 °C for 3 h. Cytoplasmic CD3 is detected when Jurkat E6 cells are phagocytosed by THP1 cells. We noticed that necrotic Jurkat E6 cells were more effectively phagocytosed by *MATR3* KD THP1 cells compared to those by control THP1 cells. *MATR3* KD THP1 cells also had a higher ability to eat necrotic Jurkat E6 cells (Figure 5 and Figure 6).

#### 3.4.3. Phagocytosis of Apoptotic Jurkat E6 Cells by shRNA-MATR3-KD THP1 and Control THP1 Cells

We treated Jurkat E6 cells at 56 °C for 30 min to induce apoptosis (Appendix A). After co-culture of apoptotic Jurkat E6 cells with THP1 cells, we noted that around 80% of apoptotic cells were up-taken by all the THP1 cell types. Although *shRNA2-MATR3*-KD and *shRNA4-MATR3* KD THP1 cells phagocytosed more apoptotic Jurkat E6 cells, compared to the control vector-transfected and naïve THP1 cells, no significant difference was observed. (Figure 5 and Figure 7A). We also cultured non-treated Jurkat E6 cells with THP1 cells for 3 h. However, there was no difference in the phagocytotic ability of *MATR3* KD cells and control THP1 cells. (Figure 5 and Figure 7B).

## 4. Discussion

HLH is a life-threatening hyperinflammatory syndrome. Primary/familial HLH is a genetic disorder that occurs mainly in young children. Secondary HLH is a hyperinflammatory condition that is triggered by numerous diseases. Infection-associated HLH accounts for approximately half of the adult HLH cases. DNA viruses, including EBV, CMV, and Herpes simplex virus, primarily contribute to infection-induced secondary HLH, whereas bacterial infections are responsible for approximately 9% of the reported adult HLH cases. Malignancies are the second leading cause of secondary HLH, accounting for approximately 15–50% of adult HLH. The most common underlying malignancies are lymphomas, especially T/NK cell lymphomas. Autoimmune diseases, including systemic juvenile idiopathic arthritis, systemic lupus erythematosus, and Kawasaki disease, also play a role in the etiology of secondary HLH [20]. Diagnosis of secondary HLH relies on clinical signs/symptoms, including fever, organomegaly, cytopenia, hypertriglyceridemia, hyperferritinemia, and elevation of serum soluble CD25 (sCD25) and soluble interleukin-2 receptor (sIL-2R) levels. However, owing to heterogeneous presentation and complex etiology, the diagnosis criteria of HLH are not well established [9,21,22,23]. Although BM hemophagocytosis is uncommon but is the hallmark of diagnosis [24]. We diagnosed HLH based on BM hemophagocytosis as well as clinical presentation, such as high fever, splenomegaly/hepatomegaly, hyperferritinemia, and hypofibrinogenemia, with/without hypertriglyceridemia. We also measured LDH levels in patients with malignancies. We did not have the resources to determine sCD25 levels at the time of diagnosis. In our limited patient sample size and etiologies, we did not find any correlation between mortality and clinical pictures, including age, sex, cytopenia, ferritin, fibrinogen, triglyceride, viral infection status, and etiologies. However, the EBV status of 38.8% of the patients was unknown. We did not observe any association between the absence of underlying diseases and the longer OS of patients.

Unlike primary/familial HLH, secondary HLH has no specific genetic background. Several gene polymorphisms related to secondary HLH have been discovered. However, some of them are relatively frequent in the general population, and several base mutations in the non-conserved gene regions are predicted to be benign in silico [20]. The immunopathogenesis of secondary HLH impairs the cytotoxic functions of NK cells and cytotoxic T lymphocytes and activates innate immune responses through stimulation of Toll-like receptors (TLR) [20]. However, Carvelli et al. reported that there was no major intrinsic cytotoxicity dysfunction, irrespective of the activated NK cell profile or interferon γ production, between patients with secondary HLH patients and controls [25].

EBV is the most common cause of secondary HLH [7,20]. Higher levels of circulating EBV DNA are associated with poor disease outcomes [26]. EBV activates NF-κB, upregulates Th1 cytokines, and leads to T cell activation in EBV-infected T cells, and thus plays a key role in the pathogenesis of HLH [12]. Bacterial and viral infections (through Toll-like receptors), inflammatory cytokines, and antigens can all lead to the activation of NF-κB, confirming its crucial role in innate and adaptive immune responses. *NF-κB* targets several genes and controls inflammatory processes and regulates apoptosis by targeting antiapoptotic Bcl family members and inhibitors of apoptosis proteins [IAPs]) and has an auto-regulatory effect on proliferation (cyclins and growth factors) [10,27,28,29]. The MATR3 gene provides instructions for making a protein called matrin 3, which is found in the nucleus of the cell as part of the nuclear matrix. The nuclear matrix is a network of proteins that provides structural support for the nucleus and aids in several important nuclear functions. The function of the matrin 3 protein is not so clear. It has been reported to bind and stabilize mRNA [30] and is a key regulator of endothelial cell survival [31]. EBV LMP-1 is the viral product responsible for the activation of the TNFR-associated NF-κB/ERK pathway to enhance cytokine secretion. There are the following two pathways that contribute to NF-κB activation: canonical and non-canonical pathways. The canonical pathway mediates inflammatory responses, and the non-canonical pathway is involved in immune cell differentiation and maturation and secondary lymphoid organogenesis [32]. Gewurz et al., reported that *MATR3* and *CGA* are the only genes that are downregulated after induction of EBV LMP1 TES2 with IκBα (a critical regulator of the transcription factor of canonical NFκB) suppressor [14]. This suggests that MATR3 may also be associated with EBV-induced HLH other than the canonical NFκB pathway. In our study, we noted that *MATR3* mRNA expression but not *NFκB* mRNA expression was downregulated in PB mononuclear cells of patients with HLH compared to that in healthy volunteers. Lower expression of *MATR3* and *NFκB* mRNA in BM cells was associated with a lower chance of survival. However, no such association was observed in PB mononuclear cells. There is no correlation between *MATR3* and *NFκB* mRNA expression between PB and BM cells. However, a positive correlation in *MATR3* and *NFκB* mRNA expression was observed in the BM cells, as well as in PB mononuclear cells. Our findings suggest that *MAT**R3* may potentially crosstalk with the NFκB pathway. Because the NFκB pathway has two pathways and *MATR3* is related to the non-canonical pathway, we only see a significant difference in *MATR3*, but not *NFκB* between HLH patients and normal volunteers. Furthermore, apart from a positive correlation between *MATR3* mRNA expression and platelet count, no correlation was observed between other clinical parameters investigated in this study.

*MATR3*mRNA expression was significantly different between patients with HLH and healthy volunteers. Therefore, we produced *MATR3*-KD THP1 cells to survey if *MATR3* expression was associated with phagocytosis. The phagocytosis amount is the highest in THP1 cells co-cultured with apoptotic Jurkat E6 cells, irrespective of *MATR3* expression. The *shRNA-MATR3* KD THP1 cells showed a higher phagocytotic effect on necrotic Jurkat E6 cells and 0.5 μL lactate beads. In our data, apoptotic cells were effectively phagocytosed, followed by necrotic cells and latex beads. Tge *shRNA-MATR3* KD THP1 cells had an enhanced phagocytic ability in necrotic cells and latex beads.

The limitations of our study are the small sample size as well as the heterogeneous etiology of HLH. Moreover, determining sCD25 and Interleukin-6 levels would have substantiated the diagnosis of HLH.

## 5. Conclusions

Secondary HLH is a hyperinflammatory disease with a high mortality rate. PB expression of *MATR3* mRNA in PB mononuclear cells of patients with HLH was significantly lower compared to that of healthy volunteers. Downregulation of *MATR3* and *NFκB* mRNA in BM cells was associated with a higher mortality rate in patients with HLH. The *shRNA-MATR3* KD THP1 cells effectively phagocytosed necrotic cells and latex beads. Here, we found a novel biomarker that has the potential to be used for the development of novel therapeutic targets.

## 6. Patents

There are no patents resulting from the work reported in this manuscript.

## Figures and Tables

**Figure 1 biomedicines-10-02161-f001:**
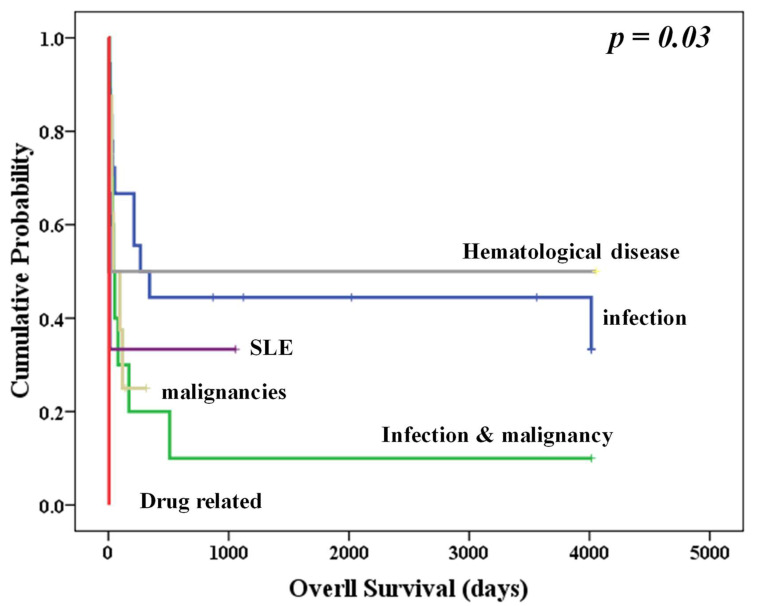
Overall survival of patients with different etiologies of secondary HLH.

**Figure 2 biomedicines-10-02161-f002:**
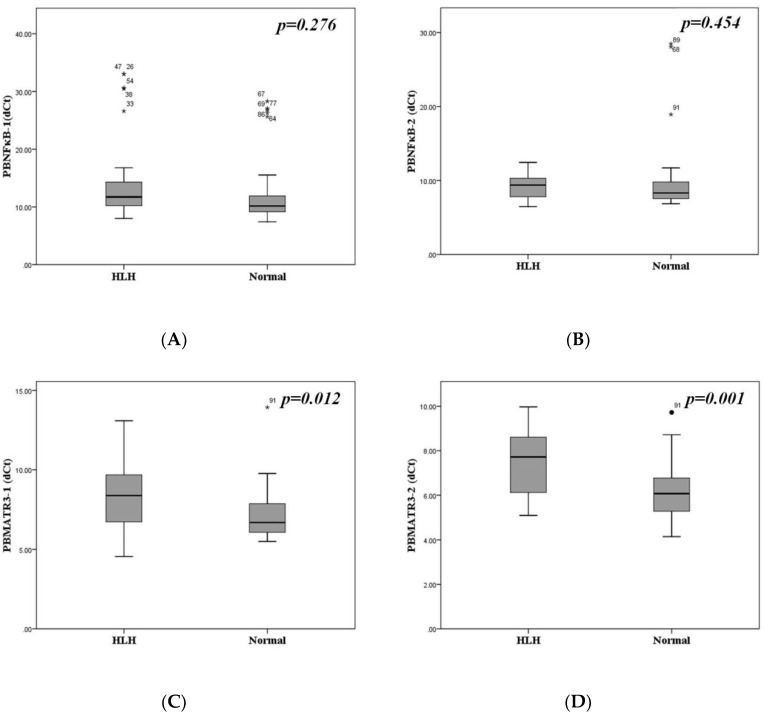
*NFκB* and *MATR3* mRNA expressions, using two different pairs of primers, in the peripheral blood mononuclear cells of patients with hemophagocytosis (HLH) and healthy volunteers. (**A**) *NFκB-1* mRNA expression (dCt), (**B**) *NFκB-2* mRNA expression (dCt), (**C**) *MATR3-1* mRNA expression (dCt), and (**D**) *MATR3-2* mRNA expression (dCt). *MATR3*-mRNA expressions showed significant difference between HLH patients and normal volunteers.

**Figure 3 biomedicines-10-02161-f003:**
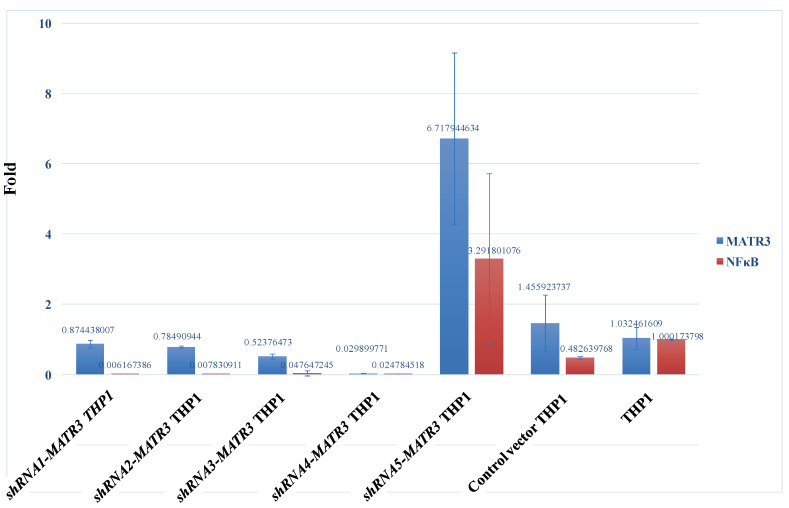
Five shRNA-*MATR3* lentivirus particles were used to knock down *MATR3* in THP1 cells. *shRNA4-MATR3* KD THP1 cells showed the lowest *MATR3* mRNA expression, followed by *shRNA3-* and *shRNA2-*
*MATR3* KD THP1 cells. There is no correlation between *MATR3* and *NF**κB* mRNA expression in *MATR3*-KD THP1 cells.

**Figure 4 biomedicines-10-02161-f004:**
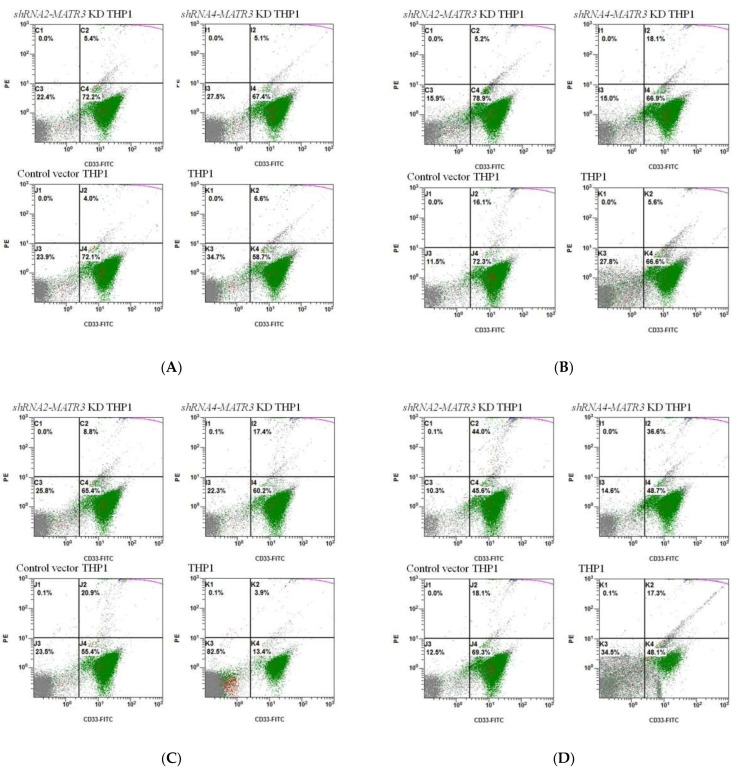
Phagocytosis of (**A**) 0.1 μL carboxylate modified polystyrene latex beads for 1 h and (**B**) 3 h, (**C**) 0.5μL carboxylate modified polystyrene latex beads for 1 h and (**D**) 3 h, in *shRNA-MATR3* KD THP1 cells. The *shRNA2-* and *shRNA4-MATR3* KD THP1 cells had a higher percentage of 0.5 μL latex beads phagocytosis in 3 h. The percentage of phagocytosis is PE(+) in THP1 cells (CD33(+) cells). (**E**) a statistic picture.

**Figure 5 biomedicines-10-02161-f005:**
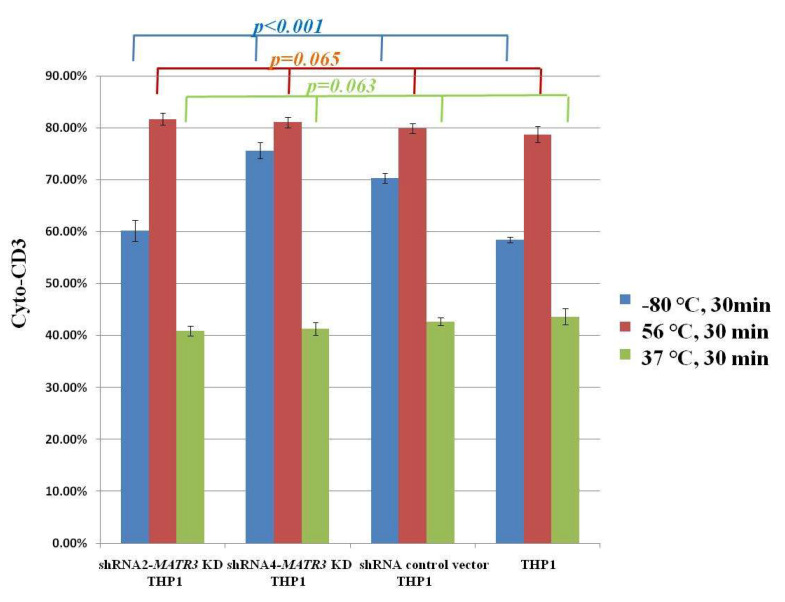
Cytoplasmic CD3 expression in *shRNA2**-*, *shRNA4**-MATR3* KD THP1, control vector transfected THP1, and naïve THP1 cells. Necrotic Jurkat E6 cells were effectively phagocytosed by *MATR3* KD THP1 cells. The y-axis represents the percentage of cytoCD3+CD13(+)/CD13(+) cells.

**Figure 6 biomedicines-10-02161-f006:**
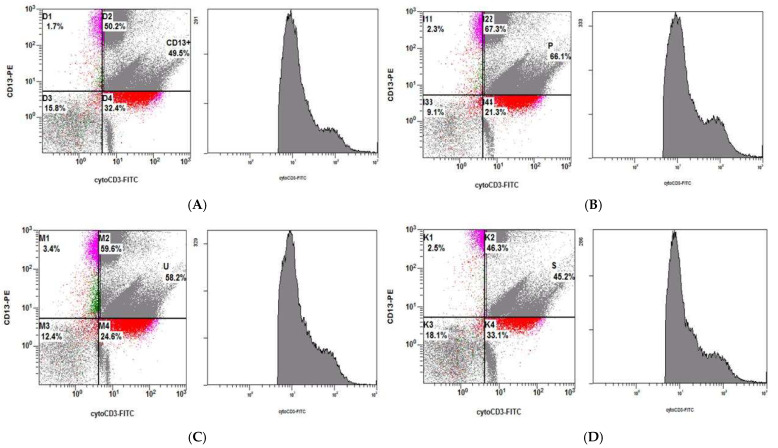
Necrotic Jurkat E6 cells were treated at −80 °C for 30 min and co-cultured with (**A**) *shRNA2-MATR3* KD THP1 cells; (**B**) *shRNA4-MATR3* KD THP1 cells; (**C**) control vector transfected THP1 cells; (**D**) naïve THP1 cells. *MATR3* KD THP1 cells had a higher ability to phagocytose necrotic Jurkat E6 cells. The y-axis represents the percentage of cytoCD3+CD13(+)/CD13(+) cells.

**Figure 7 biomedicines-10-02161-f007:**
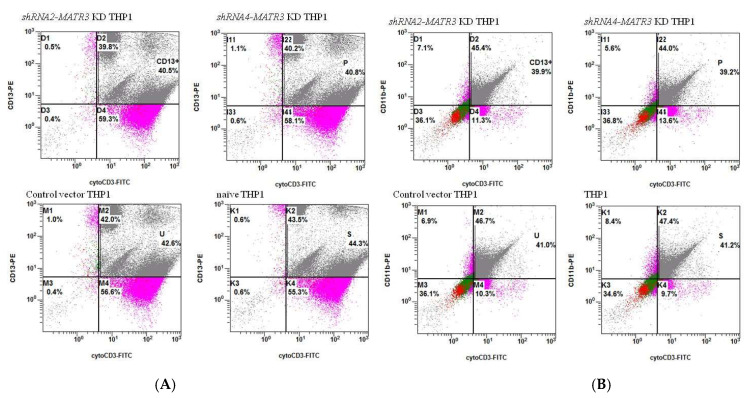
Jurkat E6 cells were treated at (**A**) 56 °C for 30 min to induce apoptosis (**B**) 37 °C for 30 min, and co-cultured with *shRNA2-MATR3* KD THP1 cells, *shRNA4-MATR3* KD THP1 cells, control vector transfected THP1 cells, and naïve THP1 cells.

**Table 1 biomedicines-10-02161-t001:** Demographic characteristics of recruited patients base on survival.

	Death (n)	Survival (n)	*p*-Value
Age (y/o) (95%CI)	51.55 (43.82–59.27) (33)	52.52 (44.09–60.96) (21)	*0.866*
Sex (male/female)	19/14	13/8	*0.489*
Ferritin (mg/dL) (95%CI)	11,847.1714 (2417.56–21,276.79) (21)	12,073.13 (−7646.19–31,792.45) (10)	*0.98*
Ferritin > 3000 ng/mL (%)	11/21 (52.38%)	3/10 (30%)	*0.218*
Triglyceride mg/dL (95%CI)	218.6 (126.59–310.61) (17)	179.7 (111.55–247.85) (10)	*0.532*
Fibrinogen (95%CI)	308.43 (225.42–391.45) (23)	354.11 (245.83–462.40) (9)	*0.523*
WBC (/μL) (95%CI)	5754.23 (2357.76–9150.70) (26)	5884.17 ± 7279.51 (1258.99–10,509.34) (12)	*0.964*
Hgb (g/dL) (95%CI)	8.996 (8.05–9.94) (26)	8.142 (7.34–8.94) (12)	*0.246*
Platelet (×1000/μL) (95%CI)	70.538 (48.25–92.83) (26)	66 (36.20–95.80) (12)	*0.807*
LDH IU/L (95%CI)	774.39 ± 996.96 (26)	406.67 ± 307.27 (12)	*0.222*
Β2-microglobulin μg/L (95%CI)	750.62 (371.71–1177.07) (13)	284.57 (211.43–601.90) (7)	*0.194*
EBV status	33	21	*0.390*
EBNA Ab (+) (%)	15 (45.45%)	13 (61.90%)	
EBV DNA (+) (%)	2 (6.06%)	4 (19.05%)	
Both negative (%)	4 (12.12%)	1 (4.76%)	
Unknown (%)	14 (42.42%)	7 (33.33%)	
CMV status	33	21	*0.661*
CMV (+) (%)	1 (3.03%)	0	
CMV (−) (%)	6 (18.18%)	3 (14.29%)	
Unknown (%)	26 (78.79%)	18 (85.71%)	
VZV status VZV (+) (%) VZV (−) (%) Unknown (%)	33	21	*0.446*
	3 (9.1%)	2 (9.52%)	
	0	1 (4.76%)	
	30 (90.91%)	18 (85.71%)	
PBNFκB-1 (dCt) (95%CI)	13.913 (10.09–17.74) (16)	16.731 (10.34–25.12) (9)	*0.092*
PBNFκB-2 (dCt) (95%CI)	9.711 (8.84–10.58) (15)	8.738 (7.46–10.01) (9)	*0.474*
PBMATR3-1 (dCt) (95%CI)	8.511 (7.71–9.31) (16)	8.491 (6.58–10.40) (9)	*0.162*
PBMATR3-2 (dCt) (95%CI)	7.751 (6.97–8.51) (16)	7.262 (6.19–8.33) (9)	*0.914*
BMNFκB-1 (dCt) (95%CI)	20.194 (16.50–23.88) (28)	15.976 (11.73–20.22) (16)	*0.011 **
BMNFκB-2 (dCt) (95%CI)	13.534 (10.78–16.29) (28)	11.946 (9.16–14.74) (17)	*0.144*
BMMATR3-1 (dCt) (95%CI)	9.382 (8.34–10.43) (30)	8.404 (7.02–9.79) (17)	*0.827*
BMMATR3-2 (dCt) (95%CI)	11.272 (8.15–14.39) (28)	8.855 (6.26–11.45) (17)	*0.017 **

PB: peripheral blood cells; BM: bone marrow; * *p* < 0.05, statistically significant; Cytomegalovirus, CMV; Varicella-Zoster Virus, VZV; lactate dehydrogenase, LDH; Epstein–Barr virus, EBV.

**Table 2 biomedicines-10-02161-t002:** Mortality and survival rates in patients with different etiologies of HLH.

Etiology	No. of Patients	Death (n = 32)	Survival (n = 18)	*p*
Infection (%)	20	12 (60%)	8 (40%)	*0.966*
Infection and malignancy (%) ^#^	9	6 (67%)	3 (33%)	
Malignancy (%) ^#^	15	10 (67%)	5 (33%)	
SLE (%)	2	1 (50%)	1 (50%)	
Vit B12 deficiency (%)	1	1 (100%)	0 (0)	
Hematological disease (%) *	2	1 (50%)	1 (50%)	
Drug-related	1	1 (100%)	0	

* Hematological disease: myelodysplastic syndrome; ^#^ malignancy: including NK/T cell and B cell lymphomas; HLH, hemophagocytic lymphohistiocytosis; SLE, systemic lupus erythematosus.

**Table 3 biomedicines-10-02161-t003:** Gene expression in different etiology of hemophagocytosis.

Cause (n)	Infection	Infection and Malignancy	Malignancies *	Hematological Diseases	SLE	Drug-Related	*p*-Value
PBNFκB-1 (dCt) (n, 95%CI)	14.916 (8.94–20.90) (10)	18.922 (6.15–31.69) (5)	16.290 (7.33–25.25) (6)	9.470 (1)	0	0	*0.746*
PBNFκB-2 (dCt) (n, 95%CI)	9.556 (8.35–10.76) (10)	9.208 (6.95–11.47) (5)	9.410 (7.56–11.266) (6)	6.720 (1)	0	0	*0.492*
PBMATR3-1 (dCt) (n, 95%CI)	8.254 (6.93–9.58) (10)	9.190 (5.81–12.57) (5)	8.162 (6.59–9.73) (6)	6.620 (1)	0	0	*0.642*
PBMATR3-2 (dCt) (n, 95%CI)	7.593 (6.39–8.79) (10)	7.632 (5.99–9.27) (5)	7.460 (6.02–8.90) (6)	5.570 (1)	0	0	*0.649*
BMNFκB-1 (dCt) (n, 95%CI)	17.584 (12.60–22.57) (15)	19.678 (5.99–9.27) (12)	21.644 (14.75–28.54) (9)	19.480 (−77.34–116.30) (2)	29.820 (1)	11.090 (1)	*0.698*
BMNFκB-2 (dCt) (n, 95%CI)	13.745 (9.91–17.58) (16)	11.304 (7.10–15.51) (12)	12.817 (8.35–17.29) (9)	18.520 (−90.4–127.54) (2)	18.40 (1)	10.680 (1)	*0.715*
BMMATR3-1 (dCt) (n, 95%CI)	9.572 (8.08–11.07) (16)	7.804 (6.54–9.06) (12)	8.850 (7.32–10.38) (9)	8.987 (5.35–12.63) (3)	8.765 (−39.07–56.60) (2)	13.640 (1)	*0.261*
BMMATR3-2 (dCt) (n, 95%CI)	11.571 (7.20–15.94) (16)	8.366 (4.42–12.31) (12)	9.622 (4.41–14.83) (9)	17.340 (−106.6–141.35) (2)	12.260 (1)	9.60 (1)	*0.683*
Ferritin (n, 95%CI)	15,807.364 (–2231.861) (14)	17,677.380 (−3803.19–39,157.95) (5)	4368.457 (−2060.83–10,797.74) (7)	247.850 (−1088.20–1583.91) (2)	20,000 (1)	7298 (1)	*0.489*
Age (n, 95%CI)	50.24 (39.87–60.60) (21)	51.44 (42.39–60.48) (16)	50.22 (32.86–67.58) (9)	73.33 (49.85–96.81) (3)	64 (2)	66 (1)	*0.855*

* Malignancies, including lymphoma; hematological diseases, including myelodysplasia syndrome and Vitamin B12 deficiency.

## Data Availability

Data supporting reported results can be obtained from corresponding author.

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
