# Peer review of "Matrin3 (MATR3) Expression Is Associated with Hemophagocytosis"

_biomedicines, 2022, doi:10.3390/biomedicines10092161_

Round 1

Reviewer 1 Report (Previous Reviewer 2)

The authors have addressed my concern in the revision. 

Author Response

reviewer: The authors have addressed my concern in the revision. 

author: Thank you very much

Reviewer 2 Report (Previous Reviewer 1)

The manuscript is interesting. However, the authors should provide the original flowcytometry plots and Fluorescent microscope about figure 4 and 5 in the manuscript. In addition, the authors should provide the activation of NFkB and expression of Matrin 3 in the THP-1 cells and the shRNA-MATR3-KD-THP1 cells. There are several typo and grammar mistakes.

Author Response

Answer to Reviewer's comment:

The manuscript is interesting. However, the authors should provide the original flowcytometry plots and Fluorescent microscope about figure 4 and 5 in the manuscript.

ANS: We add Figure 7 (original flocytometry plots). The fluorescent microscopic picture are shown in supplement Figure. It will be too massy to put everything in main text. But I provide original data in whole “set” of submission.

In addition, the authors should provide the activation of NFkB and expression of Matrin 3 in the THP-1 cells and the shRNA-MATR3-KD-THP1 cells.

Ans: Figure 3 is redrawn and add description of MATR3 and NFkB expressions’ correlation in Results part and Figure 3 legend.

There are several typo and grammar mistakes.

Ans: Thank you. I already re-confirmed and this manuscript was ever English revision by professional company “Editage”

Round 2

Reviewer 2 Report (Previous Reviewer 1)

None

This manuscript is a resubmission of an earlier submission. The following is a list of the peer review reports and author responses from that submission.

Round 1

Reviewer 1 Report

Yang et al. present the manuscript which title is Matrin3 (MATR3) expression is a novel gene associated with hemophagocytosis. The manuscript is interesting. However, there are several questions as below.

1.     In addition to MATR3 and NFkB, the authors should provide the exchange change of other mRNA in normal cases and patients.

2.     The authors should provide the number of International Review Board of the Kaohsiung Medical University Hospital in section of methods.

3.     The authors should provide the reason about the follow-up data in figure 1.

4.     The authors should provide the data in the manuscript no in the supplemental data.

5.     The authors should provide the molecular mechanism of MATR3 in the regulation of hemophagocytosis.

Reviewer 2 Report

In this study, Wen-Chi, Yang et al. investigated Matrin3 expression and its correlation with mortality of hemophagocytic lymphohistiocytosis. The author didn't justify the rationale to study this particular gene Martin3, while many genes are associated with the hyperinflammatory syndrome. With that being said, it is difficult to tell the novelty or significance of the study.  In line with this comment, figure 2 showed a mild decrease in Martin3 expression in HLH.